# DREAMFUSION: TEXT-TO-3D USING 2D DIFFUSION

**Ben Poole[1], Ajay Jain[2], Jonathan T. Barron[1], Ben Mildenhall[1]**
[1]Google Research, [2]UC Berkeley
{pooleb, barron, bmild}@google.com, ajayj@berkeley.edu

## ABSTRACT

Recent breakthroughs in text-to-image synthesis have been driven by diffusion models trained on billions of image-text pairs. Adapting this approach to 3D synthesis would require large-scale datasets of labeled 3D data and efficient architectures for denoising 3D data, neither of which currently exist. In this work, we circumvent these limitations by using a pretrained 2D text-to-image diffusion model to perform text-to-3D synthesis. We introduce a loss based on probability density distillation that enables the use of a 2D diffusion model as a prior for optimization of a parametric image generator. Using this loss in a DeepDream-like procedure, we optimize a randomly-initialized 3D model (a Neural Radiance Field, or NeRF) via gradient descent such that its 2D renderings from random angles achieve a low loss. The resulting 3D model of the given text can be viewed from any angle, relit by arbitrary illumination, or composited into any 3D environment. Our approach requires no 3D training data and no modifications to the image diffusion model, demonstrating the effectiveness of pretrained image diffusion models as priors. See dreamfusionpaper.github.io for a more immersive view into our 3D results.

## 1 INTRODUCTION

Generative image models conditioned on text now support high-fidelity, diverse and controllable image synthesis (Nichol et al., 2022; Ramesh et al., 2021; 2022; Saharia et al., 2022; 2021a; Yu et al., 2022; Saharia et al., 2021b). These quality improvements have come from large aligned image-text datasets (Schuhmann et al., 2022) and scalable generative model architectures. Diffusion models are particularly effective at learning high-quality image generators with a stable and scalable denoising objective (Ho et al., 2020; Sohl-Dickstein et al., 2015; Song et al., 2021). Applying diffusion models to other modalities has been successful, but requires large amounts of modality-specific training data (Chen et al., 2020; Ho et al., 2022; Kong et al., 2021). In this work, we develop techniques to transfer pretrained 2D image-text diffusion models to 3D object synthesis, without any 3D data (see Figure 1). Though 2D image generation is widely applicable, simulators and digital media like video games and movies demand thousands of detailed 3D assets to populate rich interactive environments. 3D assets are currently designed by hand in modeling software like Blender and Maya3D, a process requiring a great deal of time and expertise. Text-to-3D generative models could lower the barrier to entry for novices and improve the workflow of experienced artists.

3D generative models can be trained on explicit representations of structure like voxels (Wu et al., 2016; Chen et al., 2018) and point clouds (Yang et al., 2019; Cai et al., 2020; Zhou et al., 2021), but the 3D data needed is relatively scarce compared to plentiful 2D images. Our approach learns 3D structure using only a 2D diffusion model trained on images, and sidesteps this issue. GANs can learn controllable 3D generators from photographs of a single object category, by placing an adversarial loss on 2D image renderings of the output 3D object or scene (Henzler et al., 2019; Nguyen-Phuoc et al., 2019; Or-El et al., 2022). Though these approaches have yielded promising results on specific object categories such as faces, they have not yet been demonstrated to support arbitrary text.

Neural Radiance Fields, or NeRF (Mildenhall et al., 2020) are an approach towards inverse rendering in which a volumetric raytracer is combined with a neural mapping from spatial coordinates to color and volumetric density. NeRF has become a critical tool for neural inverse rendering (Tewari et al., 2022). Originally, NeRF was found to work well for "classic" 3D reconstruction tasks: many images of a scene are provided as input to a model, and a NeRF is optimized to recover the geometry of that specific scene, which allows for novel views of that scene from unobserved angles to be synthesized.

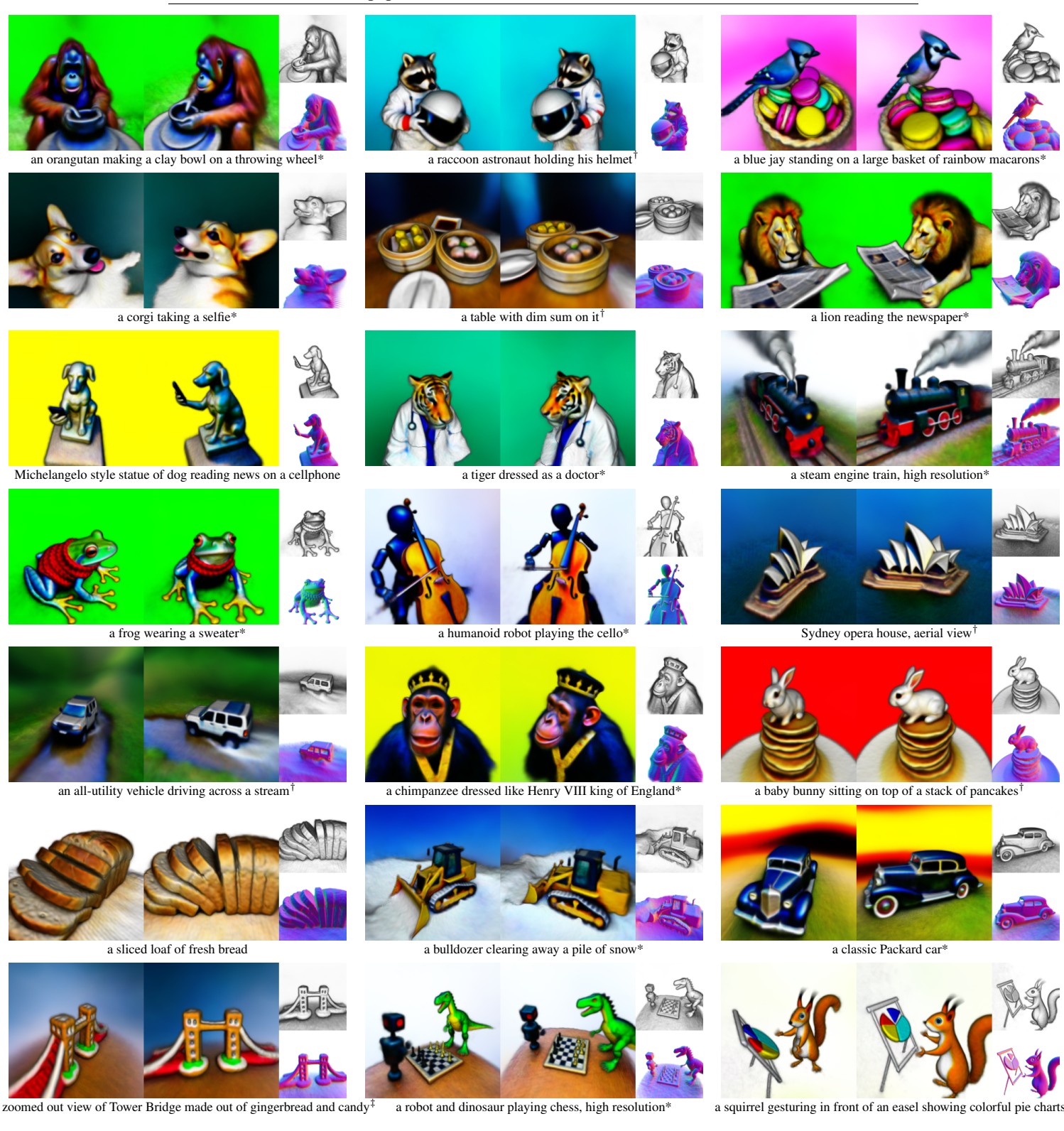

an orangutan making a clay bowl on a throwing wheel*

a raccoon astronaut holding his helmet†

a blue jay standing on a large basket of rainbow macarons*

a corgi taking a selfie*

a table with dim sum on it†

a lion reading the newspaper*

Michelangelo style statue of dog reading news on a cellphone

a tiger dressed as a doctor*

a steam engine train, high resolution*

a frog wearing a sweater*

a humanoid robot playing the cello*

Sydney opera house, aerial view†

an all-utility vehicle driving across a stream†

a chimpanzee dressed like Henry VIII king of England*

a baby bunny sitting on top of a stack of pancakes†

a sliced loaf of fresh bread

a bulldozer clearing away a pile of snow*

a classic Packard car*

zoomed out view of Tower Bridge made out of gingerbread and candy‡

a robot and dinosaur playing chess, high resolution*

a squirrel gesturing in front of an easel showing colorful pie charts

Figure 1: DreamFusion uses a pretrained text-to-image diffusion model to generate realistic 3D models from text prompts. Rendered 3D models are presented from two views, with textureless renders and normals to the right. See dreamfusionpaper.github.io for videos of these results. Symbols indicate the following prompt prefixes which we found helped to improve the quality and realism:

    * a DSLR photo of...    † a zoomed out DSLR photo of...    ‡ a wide angle zoomed out DSLR photo of...

Many 3D generative approaches have found success in incorporating NeRF-like models in the generative process (Schwarz et al., 2020; Chan et al., 2021b;a; Gu et al., 2021; Liu et al., 2022). One such approach is Dream Fields (Jain et al., 2022), which uses the frozen image-text joint embedding models from CLIP (Radford et al., 2021) and an optimization-based approach to train NeRFs. This work showed that pretrained 2D image-text models may be used for 3D synthesis, though 3D objects produced by this approach tend to lack realism and accuracy. CLIP has been used to guide other approaches based on voxel grids and meshes (Sanghi et al., 2022; Jetchev, 2021; Wang et al., 2022).

We adopt a similar approach to Dream Fields, but replace CLIP with a loss derived from distillation of a 2D diffusion model. Our loss is based on probabilty density distillation, minimizing the KL divergence between a family of Gaussian distribution with shared means based on the forward process of diffusion and the score functions learned by the pretrained diffusion model. The resulting Score Distillation Sampling (SDS) method enables sampling via optimization in differentiable image parameterizations. By combining SDS with a NeRF variant tailored to this 3D generation task, DreamFusion generates high-fidelity coherent 3D objects and scenes for a diverse set of user-provided text prompts.

## 2 DIFFUSION MODELS AND SCORE DISTILLATION SAMPLING

Diffusion models are latent-variable generative models that learn to gradually transform a sample from a tractable noise distribution towards a data distribution (Sohl-Dickstein et al., 2015; Ho et al., 2020). Diffusion models consist of a forward process $q$ that slowly removes structure from data $\mathbf{x}$ by adding noise, and a reverse process or generative model $p$ that slowly adds structure starting from noise $\mathbf{z}_t$. The forward process is typically a Gaussian distribution that transitions from the previous less noisy latent at timestep $t$ to a noisier latent at timestep $t+1$. We can compute the marginal distribution of the latent variables at timestep $t$ given an initial datapoint $\mathbf{x}$ by integrating out intermediate timesteps: $q(\mathbf{z}_t|\mathbf{x}) = \mathcal{N}(\alpha_t\mathbf{x}, \sigma_t^2\mathbf{I})$. The marginals integrating out the data density $q(\mathbf{x})$ are $q(\mathbf{z}_t) = \int q(\mathbf{z}_t|\mathbf{x})q(\mathbf{x})\,d\mathbf{x}$, and correspond to smoothed versions of the data distribution. The coefficients $\alpha_t$ and $\sigma_t$ are chosen such that $q(\mathbf{z}_t)$ is close to the data density at the start of the process ($\sigma_0 \approx 0$) and close to Gaussian at the end of the forward process ($\sigma_T \approx 1$), with $\alpha_t^2 = 1 - \sigma_t^2$ chosen to preserve variance (Kingma et al., 2021; Song et al., 2021).

The generative model $p$ is trained to slowly add structure starting from random noise $p(\mathbf{z}_T) = \mathcal{N}(\mathbf{0}, \mathbf{I})$ with transitions $p_\phi(\mathbf{z}_{t-1}|\mathbf{z}_t)$. Theoretically, with enough timesteps, the optimal reverse process step is also Gaussian and related to an optimal MSE denoiser (Sohl-Dickstein et al., 2015). Transitions are typically parameterized as $p_\phi(\mathbf{z}_{t-1}|\mathbf{z}_t) = q(\mathbf{z}_{t-1}|\mathbf{z}_t, \mathbf{x} = \hat{\mathbf{x}}_\phi(\mathbf{z}_t; t))$ where $q(\mathbf{z}_{t-1}|\mathbf{z}_t, \mathbf{x})$ is a posterior distribution derived from the forward process and $\hat{\mathbf{x}}_\phi(\mathbf{z}_t; t)$ is a learned approximation of the optimal denoiser. Instead of directly predicting $\hat{\mathbf{x}}_\phi$, Ho et al. (2020) trains an image-to-image U-Net $\epsilon_\phi(\mathbf{z}_t; t)$ that predicts the noise content of the latent $\mathbf{z}_t$: $\mathbb{E}[\mathbf{x}|\mathbf{z}_t] \approx \hat{\mathbf{x}}_\phi(\mathbf{z}_t; t) = (\mathbf{z}_t - \sigma_t\epsilon_\phi(\mathbf{z}_t; t))/\alpha_t$. The predicted noise can be related to a predicted score function for the smoothed density $\nabla_{\mathbf{z}_t} \log p(\mathbf{z}_t)$ through Tweedie's formula (Robbins, 1992): $\epsilon_\phi(\mathbf{z}_t; t) = -\sigma_t s_\phi(\mathbf{z}_t; t)$.

Training the generative model with a (weighted) evidence lower bound (ELBO) simplifies to a weighted denoising score matching objective for parameters $\phi$ (Ho et al., 2020; Kingma et al., 2021):

$$\mathcal{L}_{\text{Diff}}(\phi, \mathbf{x}) = \mathbb{E}_{t \sim \mathcal{U}(0,1), \epsilon \sim \mathcal{N}(\mathbf{0}, \mathbf{I})} \left[ w(t) \|\epsilon_\phi(\alpha_t\mathbf{x} + \sigma_t\epsilon; t) - \epsilon\|_2^2 \right], \tag{1}$$

where $w(t)$ is a weighting function that depends on the timestep $t$. Diffusion model training can thereby be viewed as either learning a latent-variable model (Sohl-Dickstein et al., 2015; Ho et al., 2020), or learning a sequence of score functions corresponding to noisier versions of the data (Vincent, 2011; Song & Ermon, 2019; Song et al., 2021). We will use $p_\phi(\mathbf{z}_t; t)$ to denote the approximate marginal distribution whose score function is given by $s_\phi(\mathbf{z}_t; t) = -\epsilon_\phi(\mathbf{z}_t; t)/\sigma_t$.

Our work builds on text-to-image diffusion models that learn $\epsilon_\phi(\mathbf{z}_t; t, y)$ conditioned on text embeddings $y$ (Saharia et al., 2022; Ramesh et al., 2022; Nichol et al., 2022). These models use classifier-free guidance (CFG, Ho & Salimans, 2022), which jointly learns an unconditional model to enable higher quality generation via a guidance scale parameter $\omega$: $\hat{\epsilon}_\phi(\mathbf{z}_t; y, t) = (1 + \omega)\epsilon_\phi(\mathbf{z}_t; y, t) - \omega\epsilon_\phi(\mathbf{z}_t; t)$. CFG alters the score function to prefer regions where the ratio of the conditional density to the unconditional density is large. In practice, setting $\omega > 0$ improves sample fidelity at the cost of diversity. We use $\hat{\epsilon}$ and $\hat{p}$ throughout to denote the guided version of the noise prediction and marginal distribution.

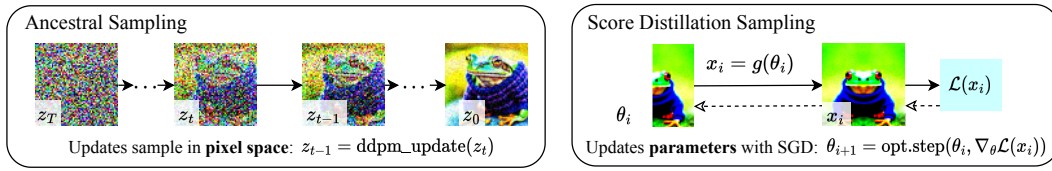

Figure 2: Comparison of 2D sampling methods from a text-to-image diffusion model with text "*a photo of a tree frog wearing a sweater.*" For score distillation sampling, as an example we use an image generator that restricts images to be symmetric by having $\mathbf{x} = (\text{flip}(\theta), \theta)$.

## 2.1 How can we sample in parameter space, not pixel space?

Existing approaches for sampling from diffusion models generate a sample that is the same type and dimensionality as the observed data the model was trained on (Song et al., 2021; 2020). Though conditional diffusion sampling enables quite a bit of flexibility (e.g. inpainting), diffusion models trained on pixels have traditionally been used to sample only pixels. We are not interested in sampling pixels; **we instead want to create 3D models that look like good images when rendered from random angles**. Such models can be specified as a differentiable image parameterization (DIP, Mordvintsev et al., 2018), where a differentiable generator $g$ transforms parameters $\theta$ to create an image $\mathbf{x} = g(\theta)$. DIPs allow us to express constraints, optimize in more compact spaces (e.g. arbitrary resolution coordinate-based MLPs), or leverage more powerful optimization algorithms for traversing pixel space. For 3D, we let $\theta$ be parameters of a 3D volume and $g$ a volumetric renderer. To learn these parameters, we require a loss function that can be applied to diffusion models.

Our approach leverages the structure of diffusion models to enable tractable sampling via optimization — a loss function that, when minimized, yields a sample. We optimize over parameters $\theta$ such that $\mathbf{x} = g(\theta)$ looks like a sample from the frozen diffusion model. To perform this optimization, we need a differentiable loss function where plausible images have low loss, and implausible images have high loss, in a similar style to DeepDream (Mordvintsev et al., 2015). We first investigated reusing the diffusion training loss (Eqn. 1) to find modes of the learned conditional density $p(\mathbf{x}|y)$. While modes of generative models in high dimensions are often far from typical samples (Nalisnick et al., 2018), the multiscale nature of diffusion model training may help to avoid these pathologies. Minimizing the diffusion training loss with respect to a **generated datapoint** $\mathbf{x} = g(\theta)$ gives $\theta^* = \arg\min_\theta \mathcal{L}_{\text{Diff}}(\phi, \mathbf{x} = g(\theta))$. In practice, we found that this loss function did not produce realistic samples even when using an identity DIP where $\mathbf{x} = \theta$. Concurrent work from Graikos et al. (2022) shows that this method can be made to work with carefully chosen timestep schedules, but we found this objective brittle and its timestep schedules challenging to tune.

To understand the difficulties of this approach, consider the gradient of $\mathcal{L}_{\text{Diff}}$:

$$\nabla_\theta \mathcal{L}_{\text{Diff}}(\phi, \mathbf{x} = g(\theta)) = \mathbb{E}_{t,\epsilon}\left[ w(t) \underbrace{(\hat{\epsilon}_\phi(\mathbf{z}_t; y, t) - \epsilon)}_{\text{Noise Residual}} \underbrace{\frac{\partial \hat{\epsilon}_\phi(\mathbf{z}_t; y, t)}{\mathbf{z}_t}}_{\text{U-Net Jacobian}} \underbrace{\frac{\partial \mathbf{x}}{\partial \theta}}_{\text{Generator Jacobian}} \right] \quad (2)$$

where we absorb the constant $\alpha_t \mathbf{I} = \partial \mathbf{z}_t / \partial \mathbf{x}$ into $w(t)$, and use the classifier-free-guided $\hat{\epsilon}_\phi$. In practice, the U-Net Jacobian term is expensive to compute (requires backpropagating through the diffusion model U-Net), and poorly conditioned for small noise levels as it is trained to approximate the scaled Hessian of the marginal density. We found that omitting the U-Net Jacobian term leads to an effective gradient for optimizing DIPs with diffusion models:

$$\nabla_\theta \mathcal{L}_{\text{SDS}}(\phi, \mathbf{x} = g(\theta)) \triangleq \mathbb{E}_{t,\epsilon}\left[ w(t) (\hat{\epsilon}_\phi(\mathbf{z}_t; y, t) - \epsilon) \frac{\partial \mathbf{x}}{\partial \theta} \right] \quad (3)$$

Intuitively, this loss perturbs $\mathbf{x}$ with a random amount of noise corresponding to the timestep $t$, and estimates an update direction that follows the score function of the diffusion model to move to a higher density region. While this gradient for learning DIPs with diffusion models may appear ad hoc, in Appendix A.4 we show that it is the gradient of a weighted probability density distillation loss (van den Oord et al., 2018) using the learned score functions from the diffusion model:

$$\nabla_\theta \mathcal{L}_{\text{SDS}}(\phi, \mathbf{x} = g(\theta)) = \nabla_\theta \mathbb{E}_t \left[ \sigma_t / \alpha_t w(t) \text{KL}(q(\mathbf{z}_t | g(\theta); y, t) \| p_\phi(\mathbf{z}_t; y, t)) \right]. \quad (4)$$

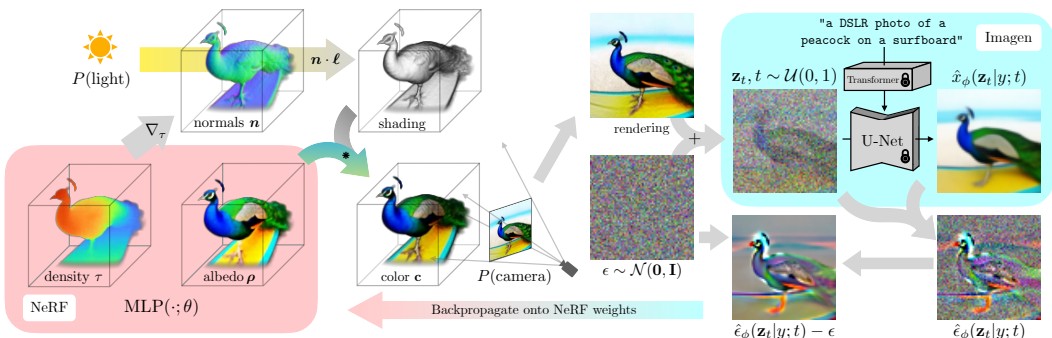

Figure 3: DreamFusion generates 3D objects from a natural language caption such as *"a DSLR photo of a peacock on a surfboard."* The scene is represented by a Neural Radiance Field that is randomly initialized and trained from scratch for each caption. Our NeRF parameterizes volumetric density and albedo (color) with an MLP. We render the NeRF from a random camera, using normals computed from gradients of the density to shade the scene with a random lighting direction. Shading reveals geometric details that are ambiguous from a single viewpoint. To compute parameter updates, DreamFusion diffuses the rendering and reconstructs it with a (frozen) conditional Imagen model to predict the injected noise $\hat{\epsilon}_\phi(\mathbf{z}_t|y;t)$. This contains structure that should improve fidelity, but is high variance. Subtracting the injected noise produces a low variance update direction $\mathrm{stopgrad}[\hat{\epsilon}_\phi - \epsilon]$ that is backpropagated through the rendering process to update the NeRF MLP parameters.

We name our sampling approach Score Distillation Sampling (SDS) as it is related to distillation, but uses score functions instead of densities. We refer to it as a sampler because the noise in the variational family $q(\mathbf{z}_t|\ldots)$ disappears as $t \to 0$ and the mean parameter of the variational distribution $g(\theta)$ becomes the sample of interest. Our loss is easy to implement (see Fig. 8), and relatively robust to the choice of weighting $w(t)$. Since the diffusion model directly predicts the update direction, **we do not need to backpropagate through the diffusion model**; the model simply acts like an efficient, frozen critic that predicts image-space edits.

Given the mode-seeking nature of $\mathcal{L}_{\mathrm{SDS}}$, it may be unclear if minimizing this loss will produce good samples. In Fig. 2, we demonstrate that SDS can generate constrained images with reasonable quality. Empirically, we found that setting the guidance weight $\omega$ to a large value for classifier-free guidance improves quality (Appendix Table 10). SDS produces detail comparable to ancestral sampling, but enables new transfer learning applications because it operates in parameter space.

## 3 THE DREAMFUSION ALGORITHM

Now that we have demonstrated how a diffusion model can be used as a loss within a generic continuous optimization problem to generate samples, we will construct our specific algorithm that allows us to generate 3D assets from text. For the diffusion model, we use the Imagen model from Saharia et al. (2022), which has been trained to synthesize images from text. We only use the $64 \times 64$ base model (not the super-resolution cascade for generating higher-resolution images), and use this pretrained model as-is with no modifications. To synthesize a scene from text, we initialize a NeRF-like model with random weights, then repeatedly render views of that NeRF from random camera positions and angles, using these renderings as the input to our score distillation loss function that wraps around Imagen. As we will demonstrate, simple gradient descent with this approach eventually results in a 3D model (parameterized as a NeRF) that resembles the text. See Fig. 3 for an overview of our approach.

### 3.1 NEURAL RENDERING OF A 3D MODEL

NeRF is a technique for neural inverse rendering that consists of a volumetric raytracer and a multilayer perceptron (MLP). Rendering an image from a NeRF is done by casting a ray for each pixel from a camera's center of projection through the pixel's location in the image plane and out into the world. Sampled 3D points $\boldsymbol{\mu}$ along each ray are then passed through an MLP, which produces 4

scalar values as output: a volumetric density $\tau$ (how opaque the scene geometry at that 3D coordinate is) and an RGB color $c$. These densities and colors are then alpha-composited from the back of the ray towards the camera, producing the final rendered RGB value for the pixel:

$$\mathbf{C} = \sum_i w_i \mathbf{c}_i , \qquad w_i = \alpha_i \prod_{j<i}(1 - \alpha_j) , \qquad \alpha_i = 1 - \exp\left(-\tau_i \left\| \boldsymbol{\mu}_i - \boldsymbol{\mu}_{i+1} \right\|\right) , \qquad (5)$$

In the traditional NeRF use-case we are given a dataset of input images and associated camera positions and the NeRF MLP is trained from random initialization using a mean squared error loss function between each pixel's rendered color and the corresponding ground-truth color from the input image. This yields a 3D model (parameterized by the weights of the MLP) that can produce realistic renderings from previously-unseen views. Our model is built upon mip-NeRF 360 (Barron et al., 2022), which is an improved version of NeRF that reduces aliasing. Though mip-NeRF 360 was originally designed for 3D reconstruction from images, its improvements are also helpful for our generative text-to-3D task (see Appendix for details).

**Shading.** Traditional NeRF models emit radiance, which is RGB color conditioned on the ray direction from which the 3D point is being observed. In contrast, our MLP parameterizes the color of the surface itself, which is then lit by an illumination that we control (a process commonly referred to as "shading"). Previous work on generative or multiview 3D reconstruction using NeRF-like models have proposed a variety of reflectance models (Bi et al., 2020; Boss et al., 2021; Srinivasan et al., 2021; Pan et al., 2022). We use an RGB albedo $\boldsymbol{\rho}$ (the color of the material) for each point:

$$(\tau, \boldsymbol{\rho}) = \mathrm{MLP}\left(\boldsymbol{\mu}; \theta\right) , \qquad (6)$$

where $\tau$ is volumetric density. Calculating the final shaded output color for the 3D point requires a normal vector indicating the local orientation of the object's geometry. This surface normal vector can be computed by normalizing the negative gradient of density $\tau$ with respect to the 3D coordinate $\boldsymbol{\mu}$: $\boldsymbol{n} = -\nabla_{\boldsymbol{\mu}}\tau / \left\| \nabla_{\boldsymbol{\mu}}\tau \right\|$ (Yariv et al., 2020; Srinivasan et al., 2021). With each normal $\boldsymbol{n}$ and material albedo $\boldsymbol{\rho}$, assuming some point light source with 3D coordinate $\boldsymbol{\ell}$ and color $\boldsymbol{\ell}_{\rho}$, and an ambient light color $\boldsymbol{\ell}_a$, we render each point along the ray using diffuse reflectance (Lambert, 1760; Ramamoorthi & Hanrahan, 2001) to produce a color $\mathbf{c}$ for each point:

$$\mathbf{c} = \boldsymbol{\rho} \circ \left(\boldsymbol{\ell}_{\rho} \circ \max\left(0, \boldsymbol{n} \cdot (\boldsymbol{\ell} - \boldsymbol{\mu}) / \left\| \boldsymbol{\ell} - \boldsymbol{\mu} \right\|\right) + \boldsymbol{\ell}_a\right) . \qquad (7)$$

With these colors and our previously-generated densities, we approximate the volume rendering integral with the same rendering weights $w_i$ used in standard NeRF (Equation 5). As in previous work on text-to-3D generation (Hong et al., 2022; Michel et al., 2021), we find it beneficial to randomly replace the albedo color $\boldsymbol{\rho}$ with white $(1, 1, 1)$ to produce a "textureless" shaded output. This prevents the model from producing a degenerate solution in which scene content is drawn onto flat geometry to satisfy the text conditioning. For example, this encourages optimization to yield a 3D squirrel instead of a flat surface containing an image of a squirrel, both of which may appear identical from certain viewing angles and illumination conditions.

**Scene Structure.** While our method can generate some complex scenes, we find that it is helpful to only query the NeRF scene representation within a fixed bounding sphere, and use an environment map generated from a second MLP that takes positionally-encoded ray direction as input to compute a background color (architecture details in Appendix A.2). We composite the rendered ray color on top of this background color using the accumulated $\alpha$ value. This prevents the NeRF model from filling up space with density very close to the camera while still allowing it to paint an appropriate color or backdrop behind the generated scene. For generating single objects instead of scenes, a reduced bounding sphere can be useful.

**Geometry regularizers.** The mip-NeRF 360 model we build upon contains many other details that we omit for brevity. We include a regularization penalty on the opacity along each ray similar to Jain et al. (2022) to prevent unneccesarily filling in of empty space. To prevent pathologies in the density field where normal vectors face backwards away from the camera we use a modified version of the orientation loss proposed in Ref-NeRF (Verbin et al., 2022). This penalty is important when including textureless shading as the density field will otherwise attempt to orient normals away from the camera so that the shading becomes darker. Full details on these regularizers and additional hyperparameters of NeRF are presented in the Appendix A.2.

### 3.2 TEXT-TO-3D SYNTHESIS

Given a pretrained text-to-image diffusion model, a differentiable image parameterization in the form of a NeRF, and a loss function whose minima are good samples, we have all the components needed

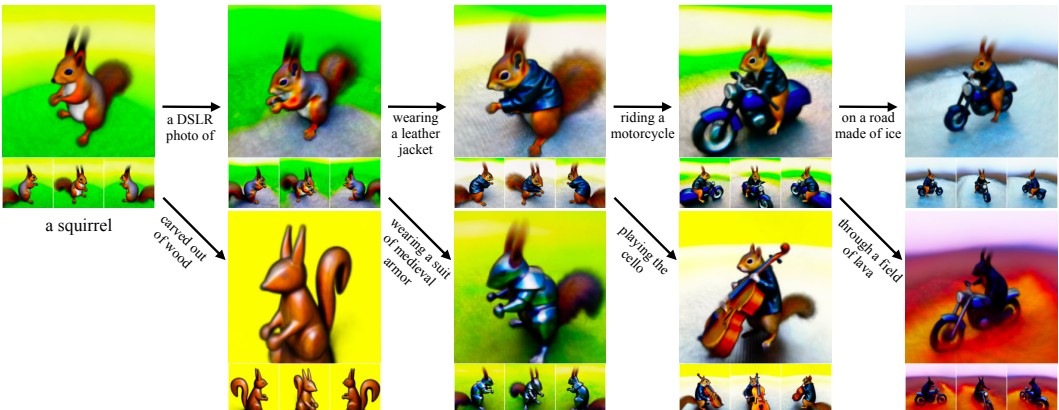

Figure 4: DreamFusion can be used to create and refine 3D scenes. Here we iteratively refine an example text prompt, while rendering each generated scene from four different viewpoints.

for text-to-3D synthesis using no 3D data. For each text prompt, we train a randomly initialized NeRF from scratch. Each iteration of DreamFusion optimization performs the following: (1) randomly sample a camera and light, (2) render an image of the NeRF from that camera and shade with the light, (3) compute gradients of the SDS loss with respect to the NeRF parameters, (4) update the NeRF parameters using an optimizer. We detail each of these steps below, and present pseudocode in Appendix 8.

**1. Random camera and light sampling.** At each iteration, a camera position is randomly sampled in spherical coordinates, with elevation angle $\phi_{\text{cam}} \in [-10°, 90°]$, azimuth angle $\theta_{\text{cam}} \in [0°, 360°]$, and distance from the origin in $[1, 1.5]$. For reference, the scene bounding sphere described previously has radius $1.4$. We also sample a "look-at" point around the origin and an "up" vector, and combine these with the camera position to create a camera pose matrix. We additionally sample a focal length multiplier $\lambda_{\text{focal}} \in \mathcal{U}(0.7, 1.35)$ such that the focal length is $\lambda_{\text{focal}}w$, where $w = 64$ is the image width in pixels. The point light position $\ell$ is sampled from a distribution centered around the camera position. We found using a wide range of camera locations to be critical for synthesizing coherent 3D scenes, and a wide range of camera distances helps to improve the resolution of the learned scene.

**2. Rendering.** Given the camera pose and light position, we render the shaded NeRF model at $64 \times 64$ resolution as described in Section 3.1. We randomly choose between the illuminated color render, a textureless render, and a rendering of the albedo without any shading.

**3. Diffusion loss with view-dependent conditioning.** Text prompts often describe canonical views of an object that are not good descriptions when sampling different views. We therefore found it beneficial to append view-dependent text to the provided input text based on the location of the randomly sampled camera. For high elevation angles $\phi_{\text{cam}} > 60°$, we append "overhead view." For $\phi_{\text{cam}} \leq 60°$, we use a weighted combination of the text embeddings for appending "front view," "side view," or "back view" depending on the value of the azimuth angle $\theta_{\text{cam}}$ (see App. A.2 for details). We use the pretrained $64 \times 64$ base text-to-image model from Saharia et al. (2022). This model was trained on large-scale web-image-text data, and is conditioned on T5-XXL text embeddings (Raffel et al., 2020). We use a weighting function of $w(t) = \sigma_t^2$, but found that a uniform weighting performed similarly. We sample $t \sim \mathcal{U}(0.02, 0.98)$, avoiding very high and low noise levels due to numerical instabilities. For classifier-free guidance, we set $\omega = 100$, finding that higher guidance weights give improved sample quality. This is much larger than image sampling methods, and is likely required due to the mode-seeking nature of our objective which results in oversmoothing at small guidance weights (see Appendix Table. 10). Given the rendered image and sampled timestep $t$, we sample noise $\epsilon$ and compute the gradient of the NeRF parameters according to Eqn. 3.

**4. Optimization.** Our 3D scenes are optimized on a TPUv4 machine with 4 chips. Each chip renders a separate view and evaluates the diffusion U-Net with per-device batch size of 1. We optimize for 15,000 iterations which takes around $1.5$ hours. Compute time is split evenly between rendering the NeRF and evaluating the diffusion model. Parameters are optimized using the Distributed Shampoo optimizer (Anil et al., 2020). See Appendix A.2 for optimization settings.

Table 1: Evaluating the coherence of DreamFusion generations with their caption using different CLIP retrieval models. We compare to the ground-truth MS-COCO images in the object-centric subset of Jain et al. (2022) as well as Khalid et al. (2022). [†]Evaluated with only 1 seed per prompt. Metrics shown in parentheses may be overfit, as the same CLIP model is used during training and eval.

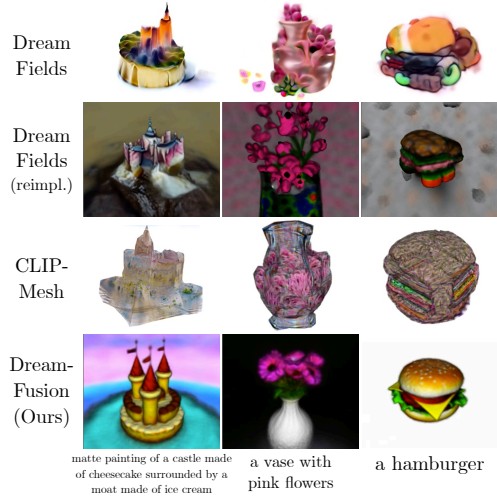

Figure 5: Qualitative comparison with baselines.

| Method | R-Precision ↑ | | | | | |
| | CLIP-B/32 | | CLIP-B/16 | | CLIP-L/14 | |
| | Color | Geo | Color | Geo | Color | Geo |
|---|---|---|---|---|---|---|
| GT Images | 77.1 | – | 79.1 | – | – | – |
| Dream Fields | 68.3 | – | 74.2 | – | – | – |
| (reimpl.) | 78.6 | 1.3 | (99.9) | (0.8) | 82.9 | 1.4 |
| CLIP-Mesh | 67.8 | – | 75.8 | – | 74.5[†] | – |
| DreamFusion (base) | **81.5** | 0.02 | **84.7** | 0.03 | **88.4** | 0.05 |
| DreamFusion | 75.1 | **42.5** | 77.5 | 46.6 | 79.7 | **58.5** |

## 4 EXPERIMENTS

We evaluate the ability of DreamFusion to generate coherent 3D scenes from a variety of text prompts. We compare to existing zero-shot text-to-3D generative models, identify the key components of our model that enable accurate 3D geometry, and explore the qualitative capabilities of DreamFusion such as the compositional generation shown in Figure 4.

3D reconstruction tasks are typically evaluated using reference-based metrics which compares recovered geometry to some ground truth. The view-synthesis literature often uses PSNR to compare rendered views with a held-out photograph. These reference-based metrics are difficult to apply to zero-shot text-to-3D generation, as there is no "true" 3D scene corresponding to our text prompts. Following Jain et al. (2022), we evaluate the CLIP R-Precision (Park et al., 2021a), an automated metric for the consistency of rendered images with respect to the input caption. The R-Precision is the accuracy with which CLIP (Radford et al., 2021) retrieves the correct caption among a set of distractors given a rendering of the scene. We use the 153 prompts from the object-centric COCO validation subset of Dream Fields. We also measure CLIP R-Precision on textureless renders to evaluate geometry since we found existing metrics do not capture the quality of the geometry, often yielding high values when texture is painted on flat geometry.

Table 1 reports CLIP R-Precision for DreamFusion and several baselines. These include Dream Fields, CLIP-Mesh (which optimizes a mesh with CLIP), DreamFusion without any view augmentations, view-dependent prompts, or shading (base) and an oracle that evaluates the original captioned image pairs in MS-COCO. We also compare against an enhanced reimplementation of Dream Fields where we use our own 3D representation (Sec. 3.1). Since this evaluation is based on CLIP, Dream Fields and CLIP-Mesh have an unfair advantage as they use CLIP during training. Despite this, DreamFusion outperforms both baselines on color images, and approaches the performance of ground truth images. While our implementation of Dream Fields performs nearly at chance when evaluating geometry (Geo) with textureless renders, DreamFusion is consistent with captions 58.5% of the time. See Appendix A.3 for more details of the experimental setup.

**Ablations.** Fig. 6 shows CLIP R-Precision for a simplified DreamFusion ablation and progressively adds in optimization choices: a large ranges of viewpoints (ViewAug), view-dependent prompts (ViewDep), optimizing illuminated renders in addition to unlit albedo color renders (Lighting), and optimizing textureless shaded geometry images (Textureless). We measure R-Precision on the albedo render as in baselines (left), the full shaded render (middle) and the textureless render (right) to check geometric quality. Geometry significantly improves with each of these choices and full renderings improve by +12.5%. Fig. 6 shows qualitative results for the ablation. This ablation also highlights how the albedo renders can be deceiving: our base model achieves the highest score, but exhibits poor geometry (the dog has multiple heads). Recovering accurate geometry requires view-dependent prompts, illumination and textureless renders.

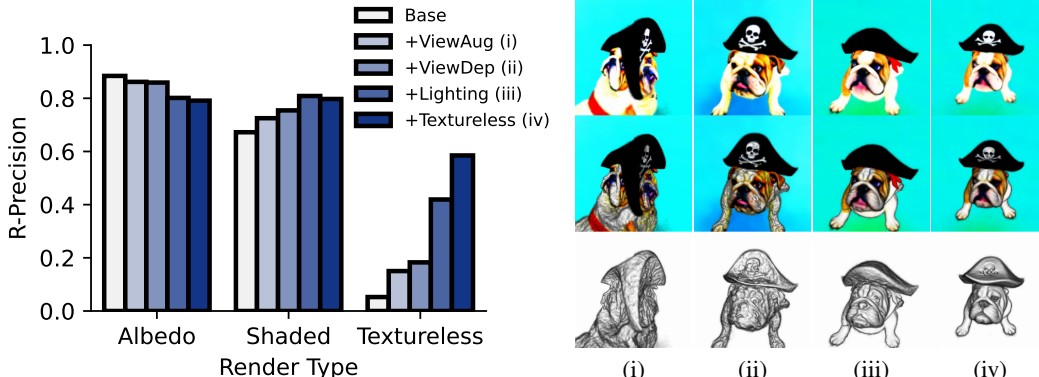

Figure 6: An ablation study of DreamFusion. **Left**: We evaluate components of our unlit renderings on albedo, full shaded and illuminated renderings and textureless illuminated geometry using CLIP L/14 on object-centric COCO. **Right**: visualizations of the impact of each ablation for *"A bulldog is wearing a black pirate hat."* on albedo (top), shaded (middle), and textureless renderings (bottom). The base method (i) without view-dependent prompts results in a multi-faced dog with flat geometry. Adding in view-dependent prompts (ii) improves geometry, but the surfaces are highly non-smooth and result in poor shaded renders. Introducing lighting (iii) improves geometry but darker areas (e.g. the hat) remain non-smooth. Rendering without color (iv) helps to smooth the geometry, but also causes some color details like the skull and crossbones to be "carved" into the geometry.

## 5 DISCUSSION

We have presented DreamFusion, an effective technique for text-to-3D synthesis for a wide range of text prompts. DreamFusion works by transferring scalable, high-quality 2D image diffusion models to the 3D domain through our use of a novel Score Distillation Sampling approach and a novel NeRF-like rendering engine. DreamFusion does not require 3D or multi-view training data, and uses only a pre-trained 2D diffusion model (trained on only 2D images) to perform 3D synthesis.

Though DreamFusion produces compelling results and outperforms prior work on this task, it still has several limitations. SDS is not a perfect loss function when applied to image sampling, and often produces oversaturated and oversmoothed results relative to ancestral sampling. While dynamic thresholding (Saharia et al., 2022) partially ameliorates this issue when applying SDS to images, it did not resolve this issue in a NeRF context. Additionally, 2D image samples produced using SDS tend to lack diversity compared to ancestral sampling, and our 3D results exhibit few differences across random seeds. This may be fundamental to our use of reverse KL divergence, which has been previously noted to have mode-seeking properties in the context of variational inference and probability density distillation.

DreamFusion uses the $64 \times 64$ Imagen model, and as such our 3D synthesized models tend to lack fine details. Using a higher-resolution diffusion model and a bigger NeRF would presumably address this, but synthesis would become impractically slow. Hopefully improvements in the efficiency of diffusion and neural rendering will enable tractable 3D synthesis at high resolution in the future.

The problem of 3D reconstruction from 2D observations is widely understood to be highly ill-posed, and this ambiguity has consequences in the context of 3D synthesis. Fundamentally, our task is hard for the same reason that inverse rendering is hard: there exist many possible 3D worlds that result in identical 2D images. The optimization landscape of our task is therefore highly non-convex, and many of the details of this work are designed specifically to sidestep these local minima. But despite our best efforts we still sometimes observe local minima, such as 3D reconstructions where all scene content is "painted" onto a single flat surface. Though the techniques presented in this work are effective, this task of "lifting" 2D observations into a 3D world is inherently ambiguous, and may benefit from more robust 3D priors.

ETHICS STATEMENT

Generative models for synthesizing images carry with them several ethical concerns, and these concerns are shared by (or perhaps exacerbated in) 3D generative models such as ours.

Because DreamFusion uses the Imagen diffusion model as a prior, it inherits any problematic biases and limitations that Imagen may have. While Imagen's dataset was partially filtered, the LAION-400M (Schuhmann et al., 2022) subset of its data was found to contain undesirable images (Birhane et al., 2021). Imagen is also conditioned on features from a pretrained large language model, which itself may have unwanted biases. It is important to be careful about the contents of datasets that are used in text-to-image and image-to-3D models so as to not propagate hateful media.

Generative models, in the hands of bad actors, could be used to generate disinformation. Disinformation in the form of 3D objects may be more convincing than 2D images (though renderings of our synthesized 3D models are less realistic than the state of the art in 2D image synthesis).

Generative models such as ours may have the potential to displace creative workers via automation. That said, these tools may also enable growth and improve accessibility for the creative industry.

REPRODUCIBILITY STATEMENT

The mip-NeRF 360 model that we build upon is publicly available through the "MultiNeRF" code repository (Mildenhall et al., 2022). While the Imagen diffusion model is not publicly available, other conditional diffusion models may produce similar results with the DreamFusion algorithm. To aid reproducibility, we have included a schematic overview of the algorithm in Figure 3, pseudocode for Score Distillation Sampling in Figure 8, hyperparameters in Appendix A.2, and additional evaluation setup details in Appendix A.3. Derivations for our loss are also included in Appendix A.4.

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

# A APPENDIX

## A.1 PSEUDOCODE FOR ANCESTRAL SAMPLING AND OUR SCORE DISTILLATION SAMPLING.

```
z_t = random.normal(img_shape)
for t in linspace(tmax, tmin, nstep):
  epshat_t = diffusion_model.epshat(z_t, y, t) # Score function evaluation.
  if t > tmin:
    eps = random.normal(img_shape)
    z_t = ddpm_update(z_t, epshat_t, eps) # 1 iteration, decreases noise level.
x = diffusion_model.xhat(z_t, epshat_t, t_min) # Tweedie's formula: denoise the last
    step.
return x
```

Figure 7: Pseudocode for ancestral sampling from DDPM where y is the optional conditioning signal e.g. a caption. Typically, $\texttt{tmax} = 1$ and $\texttt{tmin} = 1/\texttt{nstep}$. Timestep t monotonically decreases.

```
params = generator.init()
opt_state = optimizer.init(params)
diffusion_model = diffusion.load_model()
for nstep in iterations:
  t = random.uniform(0., 1.)
  alpha_t, sigma_t = diffusion_model.get_coeffs(t)
  eps = random.normal(img_shape)
  x = generator(params, <other arguments>...) # Get an image observation.
  z_t = alpha_t * x + sigma_t * eps # Diffuse observation.
  epshat_t = diffusion_model.epshat(z_t, y, t) # Score function evaluation.
  g = grad(weight(t) * dot(stopgradient[epshat_t - eps], x), params)
  params, opt_state = optimizer.update(g, opt_state) # Update params with optimizer.
return params
```

Figure 8: Pseudocode for Score Distillation Sampling with an application-specific generator that defines a differentiable mapping from parameters to images. The gradient g is computed without backpropagating through the diffusion model's U-Net. We used the stopgradient operator to express the loss, but the parameter update can also be easily computed with an explicit VJP: g = matmul(weight(t) * (-epshat_t - eps), grad(x, params)).

## A.2 NERF DETAILS AND TRAINING HYPERPARAMETERS

Our model builds upon mip-NeRF 360 (Barron et al., 2022) (starting from the publicly available implementation 2022), which is an improved version of NeRF (Mildenhall et al., 2020). The main modification this model makes to NeRF is in how 3D point information is passed to the NeRF MLP. In NeRF, each 3D input point is mapped to a higher dimensional space using a sinusoidal positional encoding function (Vaswani et al., 2017). In mip-NeRF, this is replaced by an *integrated* positional encoding that accounts for the "width" of the ray being rendered (based on its pixel footprint in the image plane) and the length of each interval $[d_i, d_{i+1}]$ sampled along the ray (Barron et al., 2021). This allows each interval along a ray to be represented as a Gaussian distribution with mean $\boldsymbol{\mu}$ and covariance matrix $\boldsymbol{\Sigma}$ that approximates the interval's 3D volume.

**Mip-NeRF covariance annealing.** As in mip-NeRF, each mean $\boldsymbol{\mu}$ is the 3D coordinate of the center of the ray interval, but unlike mip-NeRF we do not use a covariance derived from camera geometry, but instead define each $\boldsymbol{\Sigma}$ as:

$$\boldsymbol{\Sigma} = \lambda_{\boldsymbol{\Sigma}}^2 \mathbf{I}_3 \tag{8}$$

Where $\lambda_{\boldsymbol{\Sigma}}$ is a scale parameter that is linearly annealed from a large value to a small value during training. Representative settings are $5 \times 10^{-2}$ and $2 \times 10^{-3}$ for the initial and final values of $\lambda_{\boldsymbol{\Sigma}}$, linearly annealed for the first 5k steps of optimization (out of 15k total). This "coarse to fine"

annealing of a scale parameter has a similar effect as the annealing used by Park et al. (2021b) but uses integrated positional encoding instead of traditional positional encoding. The underlying sinusoidal positional encoding function uses frequencies $2^0, 2^1, \ldots, 2^{L-1}$, where we set $L = 8$.

**MLP architecture changes.**    Our NeRF MLP consists of 5 ResNet blocks (He et al., 2016) with 128 hidden units, Swish/SiLU activation (Hendrycks & Gimpel, 2016), and layer normalization (Ba et al., 2016) between blocks. We use an $\exp$ activation to produce density $\tau$ and a sigmoid activation to produce RGB albedo $\boldsymbol{\rho}$.

**Environment Map MLP.**    We use an additional MLP to create an environment map allowing the bounded NeRF MLP to be rendered on a contextually-relevant and optimized background. This MLP takes a positionally-encoded ray direction as input, and outputs an RGB color. We use positional encodings with frequencies $2^0, \ldots, 2^4$, and an MLP with 3 hidden layers of 64 units. The output of this MLP is passed through a sigmoid activation to produce RGB values between 0 and 1.

**Shading hyperparameters.**    For the first 1k steps of optimization we set the ambient light color $\boldsymbol{\ell}_a$ to $\mathbf{1}$ and the diffuse light color $\boldsymbol{\ell}_\rho$ to $\mathbf{0}$, which effectively disables diffuse shading. For the remaining steps we set $\boldsymbol{\ell}_a = [0.1, 0.1, 0.1]$ and $\boldsymbol{\ell}_\rho = [0.9, 0.9, 0.9]$ with probability 0.75, otherwise $\boldsymbol{\ell}_a = \mathbf{1}$, $\boldsymbol{\ell}_\rho = \mathbf{0}$, *i.e.* we use diffuse shading 75% of the time. When shading is on ($\boldsymbol{\ell}_\rho > 0$), we choose textureless shading ($\boldsymbol{\rho} = 1$) with probability 0.5.

**Spatial density bias.**    To aid in the early stages of optimization, we add a small "blob" of density around the origin to the output of the MLP. This helps focus scene content at the center of the 3D coordinate space, rather than directly next to the sampled cameras. We use a Gaussian PDF to parameterize the added density:

$$\tau_{\text{init}}(\boldsymbol{\mu}) = \lambda_\tau \cdot \exp\left(-\frac{\|\boldsymbol{\mu}\|^2}{2\sigma_\tau^2}\right). \tag{9}$$

Representative settings are $\lambda_\tau = 5$ for the scale parameter and $\sigma_\tau = 0.2$ for the width parameter. This density is added to the $\tau$ output of the NeRF MLP.

**Additional camera and light sampling details.**    Uniformly sampling camera elevation $\phi_{\text{cam}}$ in angular space does not produce uniform samples over the surface of the sphere — the area around the pole is oversampled. We found this bias to be helpful in practice, so we sample $\phi_{\text{cam}}$ from this biased distribution with probability 0.5, otherwise we sample from a true uniform-in-area distribution on a half-sphere. The sampled camera position is perturbed by a small uniform offset $\mathcal{U}(-0.1, 0.1)^3$. The "look-at" point is sampled from $\mathcal{N}(\mathbf{0}, 0.2\mathbf{I})$ and the default "up" vector is perturbed by noise sampled from $\mathcal{N}(\mathbf{0}, 0.02\mathbf{I})$. This noise acts as an additional augmentation, and ensures a wider diversity of viewpoints are seen during training.

We separately sample the direction and norm of the light position vector $\boldsymbol{\ell}$. To sample the direction, we sample from $\mathcal{N}(\mathbf{p}_{\text{cam}}, \mathbf{I})$ where $\mathbf{p}_{\text{cam}}$ is the camera position (this ensures that the point light usually ends up on the same side of the object as the camera). The norm $\|\boldsymbol{\ell}\|$ is sampled from $\mathcal{U}(0.8, 1.5)$, while $\|\mathbf{p}_{\text{cam}}\| \sim \mathcal{U}(1.0, 1.5)$.

**Regularizer hyperparameters.**    We use the orientation loss proposed by Ref-NeRF (Verbin et al., 2022) to encourage normal vectors of the density field to face toward the camera when they are visible (so that the camera does not observe geometry that appears to face "backwards" when shaded). We place a stop-gradient on the rendering weights $w_i$, which helps prevent unintended local minima where the generated object shrinks or disappears:

$$\mathcal{L}_{\text{orient}} = \sum_i \text{stop\_grad}(w_i) \max(0, \boldsymbol{n}_i \cdot \boldsymbol{v})^2, \tag{10}$$

where $\boldsymbol{v}$ is the direction of the ray (the viewing direction). We also apply a small regularization to the accumulated alpha value (opacity) along each ray: $\mathcal{L}_{\text{opacity}} = \sqrt{(\sum_i w_i)^2 + 0.01}$. This discourages optimization from unnecessarily filling in empty space, and improves foreground/background separation.

For orientation loss $\mathcal{L}_{\text{orient}}$, we find reasonable weights to lie in $[10^{-1}, 10^{-3}]$. If orientation loss is too high, surfaces become oversmoothed. In most experiments, we set the weight to $10^{-2}$. This weight

is annealed in starting from $10^{-4}$ over the first 5k (out of 15k) steps. For accumulated alpha loss $\mathcal{L}_{\text{opacity}}$, we find reasonable weights to lie in $[10^{-3}, 5 \times 10^{-3}]$.

**View-dependent prompting.**    We interpolate between front/side/back view prompt augmentations based on which quadrant contains the sampled azimuth $\theta_{\text{cam}}$. We experimented with different ways of interpolating the text embeddings, but found that simply taking the text embedding closest to the sampled azimuth worked well.

**Optimizer.**    We use Distributed Shampoo (Anil et al., 2020) with $\beta_1 = 0.9$, $\beta_2 = 0.9$, `exponent_override` $= 2$, `block_size` $= 128$, `graft_type` $=$ `SQRT_N`, $\epsilon = 10^{-6}$, and a linear warmup of learning rate over 3000 steps from $10^{-9}$ to $10^{-4}$ followed by cosine decay down to $10^{-6}$. We found this long warmup period to be helpful for improving the coherence of generated geometry.

## A.3    EXPERIMENTAL SETUP

Our computation of R-Precision differs slightly from baselines. As mentioned, CLIP-based text-to-3D systems are prone to overfitting the evaluation CLIP R-Precision metric since the model used for training is similar to the evaluation model. To minimize this problem, Dream Fields (Jain et al., 2022) and CLIP-Mesh (Khalid et al., 2022) evaluate renderings at a single held out view at a $45°$ elevation, higher than is seen during training (maximum $30°$). DreamFusion evaluates at $30°$ since it is not prone to this issue, but averages the metric over multiple azimuths to reduce variance. In our main results in Table 1, we evaluate all captions with 2 generation seeds unless otherwise noted.

## A.4    DERIVING THE SCORE DISTILLATION SAMPLING LOSS AND GRADIENTS

The score distillation sampling loss $\mathcal{L}_{\text{SDS}}$ presented in Eqn. 4 was inspired by work on probability density distillation (Huang et al., 2019; Ping et al., 2019; van den Oord et al., 2018). We use this loss to find modes of the score functions that are present across all noise levels in the diffusion process. Here we show how the gradient of this loss leads to the same update as optimizing the training loss $\mathcal{L}_{\text{Diff}}$, but without the term corresponding to the Jacobian of the diffusion U-Net. First, we consider gradients with respect to a single KL term:

$$\text{KL}(q(\mathbf{z}_t|\mathbf{x} = g(\theta))\|p_\phi(\mathbf{z}_t|y)) = \mathbb{E}_\epsilon \left[\log q(\mathbf{z}_t|\mathbf{x} = g(\theta)) - \log p_\phi(\mathbf{z}_t|y)\right] \tag{11}$$

$$\nabla_\theta \text{KL}(q(\mathbf{z}_t|\mathbf{x} = g(\theta))\|p_\phi(\mathbf{z}_t|y)) = \mathbb{E}_\epsilon \left[\underbrace{\nabla_\theta \log q(\mathbf{z}_t|\mathbf{x} = g(\theta))}_{(A)} - \underbrace{\nabla_\theta \log p_\phi(\mathbf{z}_t|y)}_{(B)}\right] \tag{12}$$

The second term (B) can be related to $\hat{\epsilon}$ by the chain rule, relying on $s_\phi(\mathbf{z}_t|y) \approx \nabla_{\mathbf{z}_t} \log p_\phi(\mathbf{z}_t|y)$:

$$\nabla_\theta \log p_\phi(\mathbf{z}_t|y) = s_\phi(\mathbf{z}_t|y)\frac{\partial \mathbf{z}_t}{\partial \theta} = \alpha_t s_\phi(\mathbf{z}_t|y)\frac{\partial \mathbf{x}}{\partial \theta} = -\frac{\alpha_t}{\sigma_t}\hat{\epsilon}_\phi(\mathbf{z}_t|y)\frac{\partial \mathbf{x}}{\partial \theta} \tag{13}$$

The first term (A) is the gradient of the entropy of the forward process with respect to the mean parameter, holding the variance fixed. Because of the fixed variance, the entropy is constant for a given $t$ and the total gradient of (A) is 0. However, we can still write out its gradient in terms of the "score function" (the gradient of the log probability with respect to parameters) and path derivative (the gradient of the log probability with respect to the sample):

$$\nabla_\theta \log q(\mathbf{z}_t|\mathbf{x}) = \Big(\underbrace{\frac{\partial \log q(\mathbf{z}_t|\mathbf{x})}{\partial \mathbf{x}}}_{\text{parameter score}} + \underbrace{\frac{\partial \log q(\mathbf{z}_t|\mathbf{x})}{\partial \mathbf{z}_t}\frac{\partial \mathbf{z}_t}{\partial \mathbf{x}}}_{\text{path derivative}}\Big)\alpha_t\frac{\partial \mathbf{x}}{\partial \theta} = \Big(\frac{\alpha_t}{\sigma_t}\epsilon - \frac{\alpha_t}{\sigma_t}\epsilon\Big)\alpha_t\frac{\partial \mathbf{x}}{\partial \theta} = 0. \tag{14}$$

Sticking-the-Landing (Roeder et al., 2017) shows that keeping the path derivative gradient while discarding the score function gradient can lead to reduced variance as the path derivative term can be correlated with other terms in the loss. Here, the other term (B) corresponds to a prediction of $\epsilon$, which is certainly correlated with the RHS of equation 14. Putting these together, we can use a "sticking-the-landing"-style gradient of our loss by thinking of $\epsilon$ as a control variate for $\hat{\epsilon}$:

$$\nabla_\theta \mathcal{L}_{\text{SDS}} = \mathbb{E}_{t,\mathbf{z}_t|\mathbf{x}}\left[w(t)\frac{\sigma_t}{\alpha_t}\nabla_\theta \text{KL}(q(\mathbf{z}_t|\mathbf{x} = g(\theta))\|p_\phi(\mathbf{z}_t|y))\right] \tag{15}$$

$$= \mathbb{E}_{t,\epsilon}\left[w(t)\left(\hat{\epsilon}(\mathbf{z}_t|y) - \epsilon\right)\frac{\partial \mathbf{x}}{\partial \theta}\right]. \tag{16}$$

In practice, we find that including $\epsilon$ in the gradient leads to lower-variance gradients that speed up optimization and can produce better final results.

In related work, Graikos et al. (2022) also sample diffusion models by optimization, thereby allowing parameterized samples. Their divergence $\mathrm{KL}(h(\mathbf{x}) \parallel p_\phi(\mathbf{x}|y))$ reduces to the loss $\mathbb{E}_{\epsilon,t}\|\epsilon - \hat{\epsilon}_\theta(\mathbf{z}_t|y;t)\|_2^2 - \log c(\mathbf{x}, y)$. The squared error requires costly backpropagation through the diffusion model $\hat{\epsilon}_\theta$, unlike SDS. DDPM-PnP also uses an auxiliary classifier $c$, while we use CFG.

A few other works have updates resembling Score Distillation Sampling for different applications. Gradients of the entropy of an implicit model have been estimated with an amortized score model at a single noise level (Lim et al., 2020), though that work does not use our control variate based on subtracting the noise from $\hat{\epsilon}$. SDS could also ease optimization by using multiple noise levels. GAN-like amortized samplers can be learned by minimizing the Stein discrepancy (Hu et al., 2018; Grathwohl et al., 2020), where the optimal critic resembles the difference of scores in our loss (Eqn. 3).

## A.5    IMPACT OF SEED AND GUIDANCE WEIGHT

We find that large guidance weights are important for learning high-quality 3D models. Unlike image synthesis models that use guidance weights $\omega \in [5, 30]$, DreamFusion uses weights up $\omega = 100$, and works at even larger guidance scales without severe artifacts. This may be due to the constrained nature of the optimization problem: colors output by our MLP are bounded to $[0, 1]$ by a sigmoid nonlinearity, whereas image samplers need clipping.

We also find that our method does not yield large amounts of diversity across random seeds. This is likely due to the mode-seeking properties of $\mathcal{L}_{\mathrm{SDS}}$ combined with the fact that at high noise levels, the smoothed densities may not have many distinct modes. Understanding the interplay between guidance strength, diversity, and loss functions remains an important open direction for future research.

## A.6    EXTENDED QUALITATIVE COMPARISON WITH PRIOR WORK

In Figure 11, we compare DreamFusion to Dream Fields on several prompts from the object-centric MS-COCO validation set.

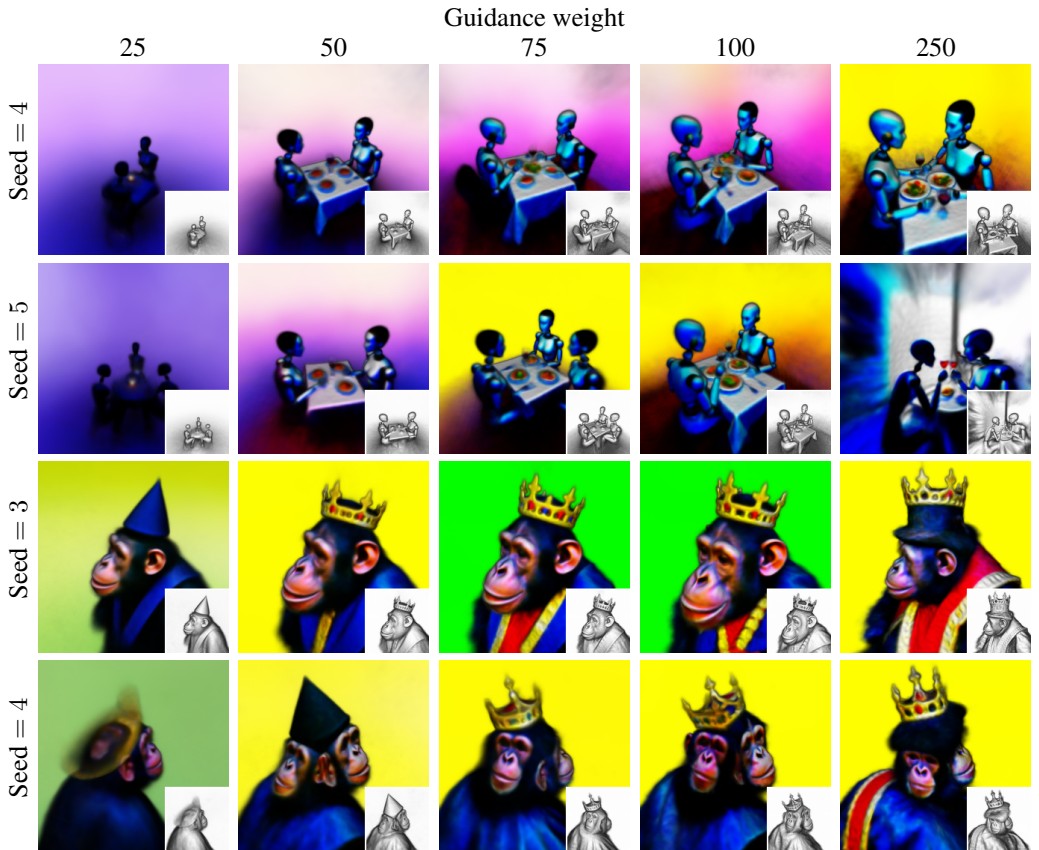

Figure 9: A 2D sweep over guidance weights and random seeds for two different prompts ("*a zoomed out DSLR photo of a robot couple fine dining*" and "*a DSLR photo of a chimpanzee dressed like Henry VIII king of England*").

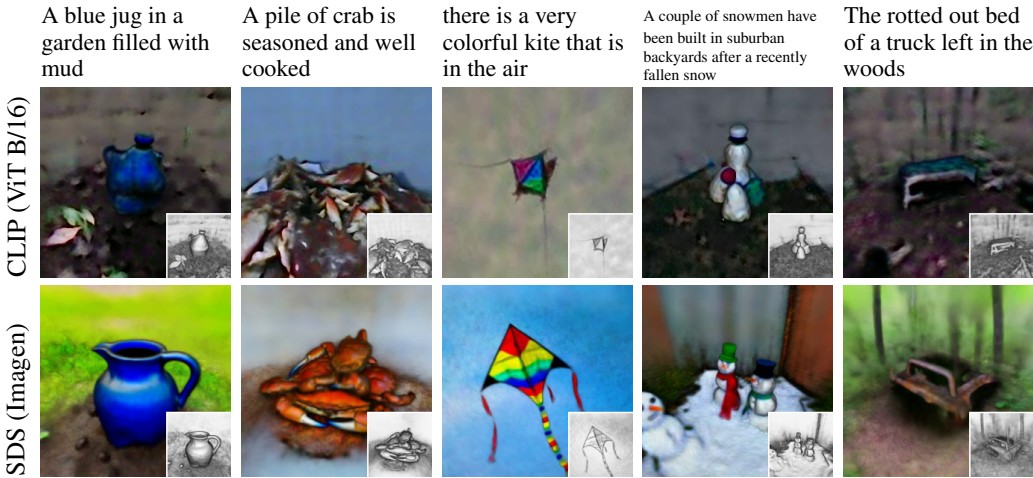

Figure 10: Comparison of the CLIP loss from Dream Fields (top) and our SDS loss. For fair comparison of the loss functions in isolation, we use all of our proposed methods including view-dependent prompts, shading, optimizing untextured renderings, and regularizers, with the same 3D NeRF representation. Qualitatively, 3D scenes generated with SDS (ours) are much more coherent than scenes generated with a CLIP loss.

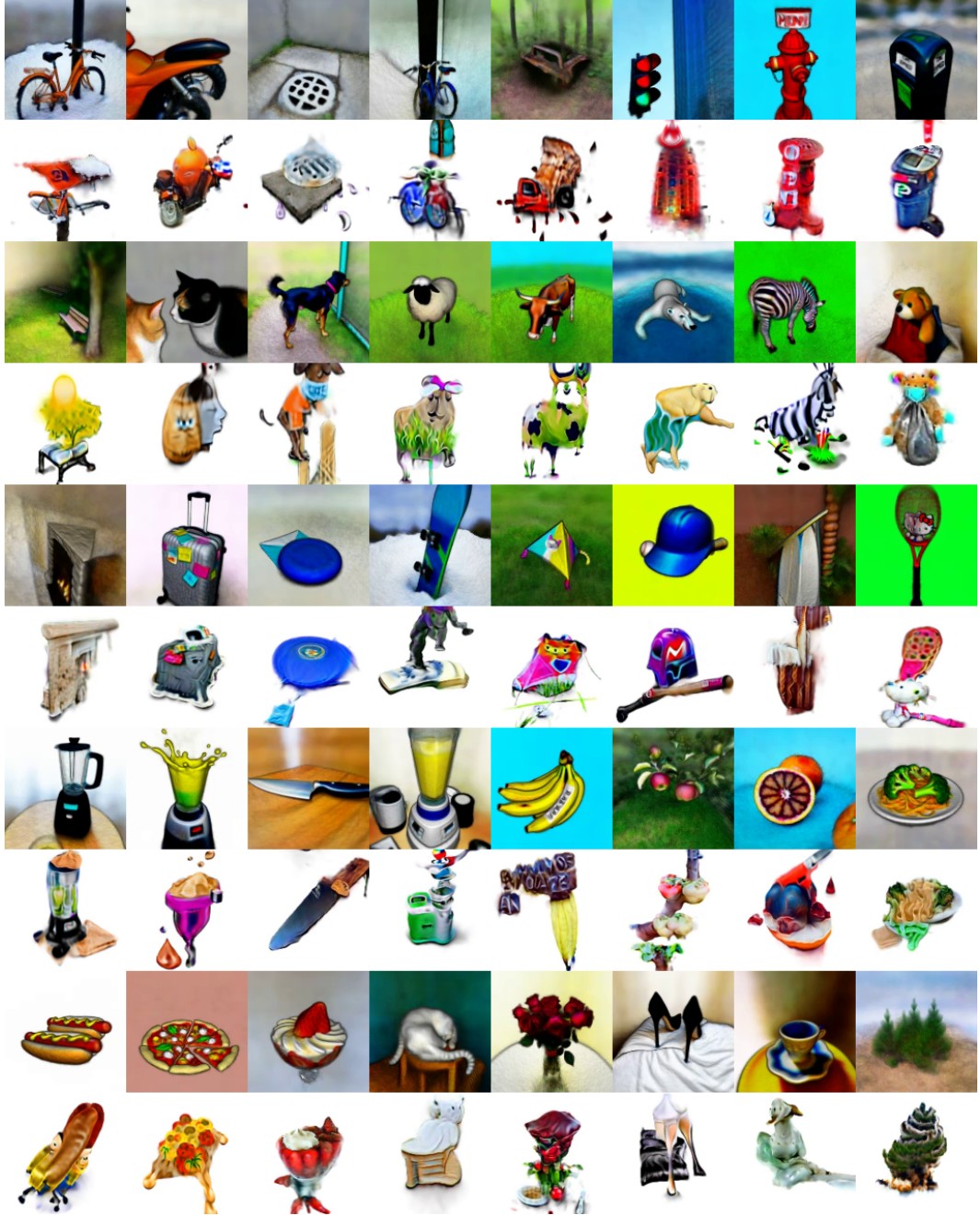

Figure 11: Comparison between DreamFusion and Dream Fields (Jain et al., 2022) on object-centric MS-COCO prompts. Odd rows are results from DreamFusion, and even rows are results from Dream Fields. DreamFusion exhibits increased global coherence and quality.

