# OpenReview forum: "DreamFusion: Text-to-3D using 2D Diffusion"
_ICLR.cc/2023/Conference — ICLR 2023 notable top 5%_

### Official Review · Reviewer_nnv7 · 2022-10-24

**Confidence:** 4
**Correctness:** 3
**Technical Novelty And Significance:** 3
**Empirical Novelty And Significance:** 3
**Recommendation:** 8

**Clarity, Quality, Novelty And Reproducibility:**

As described in weaknesses, I believe the paper needs some effort in clarifying the differences between DreamFusion and Dream Fields.

**Strength And Weaknesses:**

Strengths:
1. The visual quality is obviously phenomenal in generative modeling research, although the formulation is not generative modeling. Under the umbrella of creative content synthesis, it becomes interesting whether do we really need a real-time generative model, or simply a faster (1.5 hours in this work is still too long) inversion method.

Weaknesses:
1. This is personal, but I do not like the writing strategy of the paper. Many of the core concepts in this paper are borrowed from Dream Fields, but the abstract and introduction are written in a way as if the concept is first proposed in this paper. And the differentiation between DreamFusion and Dream Fields is written in obscure sentences. I would suggest the authors take a deeper look into how to present the differences.
2. In Figure 5., the re-implementation version of Dream Fields is visually much worse than the original version, despite the quantitative scores seeming to be fine. I would suggest the authors adding more visual comparisons between these two methods in the Appendix.

Questions:
1. Current 3D generative models mostly rely on a canonical coordinate to align objects, and therefore fail on categories without aligned canonical coordinates (such as trees or scenes). Does Dream Fusion have a similar limitation?

**Summary Of The Paper:**

The paper uses a NeRF backbone with CLIP-based text guidance to synthesize 3D objects. Although most components are borrowed from its prior work, Dream Fields, the paper focuses on the sampling algorithm, and shows superior performance on both qualitative and quantitative results.

**Summary Of The Review:**

This paper is a clear acceptance, but there are some minor issues in the writing. And I see the TPUs are used in this paper, despite unlikely, but it would be greatly appreciated if the authors can make the codes publicly available soon, as an effort towards an open and welcoming research community.

---

> ### Author Response · Authors · 2022-11-10
> **Response to Reviewer nnv7**
>
> Thank you for taking the time to read and engage with our submission. We respond below to your questions and concerns:
>
> > The paper uses a NeRF backbone with CLIP-based text guidance to synthesize 3D objects.
>
> Our work (unlike Dream Fields) does not use CLIP-based guidance. We use a text-to-image diffusion model to guide the synthesis of 3D objects.
>
>
> > Framing relative to Dream Fields.
>
> While the core idea of randomly rendering a NeRF and scoring it with a model trained on image-text data is shared with Dream Fields, we have introduced several new components that greatly improve the quality:
> - Proposed a new loss function to synthesize from diffusion models via optimization (Section 2)
> - Introduced illumination and textureless rendering to shade the NeRF and greatly improve the quality of geometry (see the CLIP R-Precision Geo scores in Table 1)
> - Increased the range of views and added view-dependent prompts to generate objects that are not radially symmetric
> - Demonstrated empirically and qualitatively that these advances enable generation of realistic 3D objects from diverse text prompts
>
> We take great inspiration from Dream Fields, and in the introduction say “We adopt a similar approach to Dream Fields” and cite this work throughout.
>
>
> > worse visual quality of Dream Fields reimplementation
>
> We agree that our reimplementation of Dream Fields produces different looking results than those presented in the Dream Fields paper, but we are uncertain if one implementation obviously produces qualitatively better results than the other --- both results seem somewhat poor (and our reimplementation’s result for `a vase with pink flowers` in Figure 5 arguably looks significantly better than the original implementation’s result). Differences in our reimplementation include an improved NeRF model (mip-NeRF 360) and an environment map for the background. We have included a more extensive qualitative comparison between Dream Fields and DreamFusion in Figure 10 of our new Appendix A.6.
>
>
> > canonical coordinates
>
> We do use view-dependent prompts to describe front/side/back of objects, which results in a somewhat canonical orientation. However, our method performs equally well on radially symmetric objects (e.g. `dim sum` in Fig 1) or objects without a canonical orientation (e.g. `hamburger` in Fig 5).

---

### Official Review · Reviewer_Ccog · 2022-10-24

**Confidence:** 3
**Correctness:** 3
**Technical Novelty And Significance:** 4
**Empirical Novelty And Significance:** Not applicable
**Recommendation:** 8

**Clarity, Quality, Novelty And Reproducibility:**

Clarity: The method is described clearly, while some variables or operations are not well defined in the main text.

The quality and novelty are very good, it is a big step to generate 3D models from text without 3D labeling.

The reproducibility is possible with diffusion models other than Imagen, but may not be totally reproducing the results due to different diffusion model qualities.



**Details Of Ethics Concerns:**

Users may generate unpleasant 3D content with the proposed method.

**Strength And Weaknesses:**

++ The proposed method should be useful for 3D asset generation, and the results are very promising.

++ The loss from Score Distillation Sampling opens a new way to use the 2D diffusion model as prior for NerF model optimization.

++ The diffusion model is pretrained and can be directly used in the text-to-3D task.

There are also several concerns.
-- In Fig. 1, some images are without a symbol indicating the prefix, and there is no "wide angle zoomed out..."

-- CLIP model is not explained and cited in Sec. 1 while Sec. 4 shows the reference of CLIP.

-- The ground truth noise $\epsilon$ in Eq. 1 is not defined or explained before use. What does $\epsilon$ look like?

-- It is better to define the KL operation in the main text near Eq. 4, letting the reader have a smoother understanding.

-- Why use Imagen in the proposed method?  How is the performance if using other diffusion models?



**Summary Of The Paper:**

This paper presents a novel method to generate 3D assets from text descriptions.
The proposed method utilizes a pretrained diffusion model to generate a denoised 2D image from text and noisy input.
The noisy input is rendered with a NerF model and known noise. The NerF model is trained by minimizing the noise residual between known noise and predicted noise.
The experimental results and the galleries in the supplementary material show the proposed method works quite well.


**Summary Of The Review:**

The paper presents a novel solution for 3D assets generation from text, and the results show the proposed method work very well.
Some descriptions can be improved to make readers a smoother understanding.

---

> ### Author Response · Authors · 2022-11-10
> **Response to Reviewer Ccog**
>
> Thank you for the careful reading and helpful feedback. We respond below to your questions and concerns:
>
>
> > Fig 1 caption
>
> Panels in Figure 1 without a symbol have no additional prefix. We removed the caption “wide angle zoomed out…”  as none of the examples in Fig 1 use this prefix.
>
>
> > CLIP references
>
> We have added a citation where CLIP is first mentioned in Section 1, and a citation to Park et al. (2021) for CLIP R-Precision when it is first mentioned. We use these models as-is, so we believe referencing this prior work is sufficient, but we are happy to add an Appendix section detailing the exact calculation of these metrics if the reviewers think it would be helpful.
>
>
> > $\epsilon$ in Eqn 1
>
> $\epsilon$ is defined in Equation 1 under the expectation ($\epsilon \sim \mathcal{N}(0, 1)$), it’s an iid Gaussian vector with mean 0, covariance I, and dimensionality the same as the image $x$
>
>
> > KL definition
>
> Are you requesting a definition of KL divergence, or the derivation linking Eqn 3 and 4 in particular? If the latter, that derivation can be found in Appendix A.4. If the former, we are happy to reproduce the definition of KL divergence in the text if the reviewers agree it is necessary.
>
>
> > use of Imagen
>
> We used Imagen because it was the best-performing text-to-image model that we had access to. The question of how well this model would perform with other text-to-image diffusion models is a great one, and it currently appears to be an open question. The previously-mentioned Stable Diffusion reimplementation of our work currently produces worse results than our implementation, which we think may be due to Stable Diffusion having a reduced sensitivity to view-dependent prompts. In the future we would love to better understand how the quality of the text-to-image model impacts the quality of 3D generation.

---

> > ### Comment · Reviewer_Ccog · 2022-12-07
> > **Response to the authors' response**
> >
> > Thank you for the authors' response and the efforts to improve the manuscript.
> > I wish to see the following work to deeper understand the relationship between text-to-image models and 3D generation, and provide the open-source model for this task.

---

### Official Review · Reviewer_UY2y · 2022-10-27

**Confidence:** 4
**Correctness:** 4
**Technical Novelty And Significance:** 4
**Empirical Novelty And Significance:** 4
**Recommendation:** 8

**Clarity, Quality, Novelty And Reproducibility:**

The paper is well-written. The algorithm is novel. See more comments in Strength And Weaknesses.

**Strength And Weaknesses:**

Strength:

+ The method allows flexible control of the resulting 3D model of the given text, e.g. it can be viewed from any angle, relit by arbitrary illumination, or composited into any 3D environment.

+ The approach requires no 3D training data and no modifications to the image diffusion model.

+ The 3D rendering process is well-considered such that many local minimums can be avoided.

+ The paper is well written, and one can quickly grasp the main idea and many technical details.

+ The paper presents a novel way to sample parameter space instead of pixel space.


Weakness:

- On P4, the paper says that it minimizes the diffusion training loss: $L_{Diff}$. As shown in Eq. 1, it is $\epsilon_\phi$ that is used to compare with $\epsilon$. However, in Eq. 2, the variable becomes $\hat{\epsilon}_\phi$. Please explain the discrepancy here.

- On P6, above Eq. 5, the paper says that it performs alpha-composition from the back of the ray to the camera. This is quite unusual, as in NeRF we usually go the opposite way, i.e. from near camera to far, such that objects near the camera are first added, and the further objects can be occluded.

- There seems no specular term in Eq. 7. However, some specular effects can be seen in Figure 4, e.g. on the armor. I wonder where those effects come from.

- On P6, the description of the second MLP for scene generation is too brief. How important is it for the final synthesis performance? Why does this MLP only take ray direction as input? Why does it only produce RGB color instead of 3D geometry as well?

- Some important results are missing from the ablation study: 1) instead of adding components one by one, the study should consider removing each component separately, such that it is easier to understand which component is the most relevant; 2) the comparison between the new SDS loss and the previous CLIP loss in DreamField is missing.

- Minor issue: In Table 1, it is better to add a dash in the CLIP names, for example, CLIP-B/32.

**Summary Of The Paper:**

This paper presents a Text-to-3D algorithm by using a pre-trained 2D text-to-image diffusion model.
It modifies the denoising diffusion loss for better supervision signal.
It also introduces a probability explanation of the loss called density distillation.
Using this loss in a DeepDream-like procedure, the method optimizes a randomly-initialized NeRF via gradient descent such that its 2D renderings from random angles achieve a low loss.

**Summary Of The Review:**

The paper is well-written. The algorithm is novel. See more comments in Strength And Weaknesses.

---

> ### Author Response · Authors · 2022-11-10
> **Resonse to Reviewer UY2y**
>
> Thank you for the careful reading and helpful feedback.  We respond below to your questions and concerns:
>
> > $\epsilon$ vs. $\hat\epsilon$
>
> $\hat\epsilon$ corresponds to the classifier-free-guided epsilon prediction (see bottom of page 3). We have clarified this in the text.
>
>
> > “front to back” vs “back to front” alpha-composition
>
> Indeed, the way that alpha composition is often described can be somewhat ambiguous! The reviewer is indeed correct that the math for implementing alpha composition is performed “front to back”, usually with a cumulative sum or cumulative product. However, alpha composition can be accurately described in either direction; for example, the painter’s algorithm (https://en.wikipedia.org/wiki/Painter%27s_algorithm) is “back to front”, but Z-buffering / Z-culling (https://en.wikipedia.org/wiki/Z-buffering) is “front to back”. We used the “back to front” description, but this is arbitrary, and we are happy to omit or invert this description if the reviewers find it more confusing than informative.
>
>
> > specular term
>
> Though our rendering model does not explicitly model specularities, NeRF is able to
> fake specular effects. Often this is as simple as just “painting” a whiter spot onto the surface of an object, but sometimes NeRF may cleverly embed a spot of bright radiance within a semi-transparent dark surface such that the bright spot is only visible from some angles. This effect has been addressed in the NeRF literature (see https://dorverbin.github.io/refnerf/) but it is currently unclear how these advances in NeRF can be used within 3D generative modeling.
>
>
> > second MLP for scene generation
>
> The additional MLP is to create an environment map/background assumed to exist at infinite distance from the origin, which is why it only takes ray direction and does not produce density or geometry. Optimizing this environment map separately while constraining the main NeRF to lie within a bounding sphere lets us synthesize isolated objects, but prevents us from synthesizing richer scenes as full context. This background MLP does not contribute much to the quality of the synthesized object itself, but rather enables better-looking renderings by placing the central object within a contextually-relevant background (e.g. placing an outdoor object under a blue sky). We have added more discussion and details of the environment map MLP to Appendix A.2.
>
>
> > missing ablations on components
>
> Thank you for the suggestion to directly compare the SDS loss with CLIP. We have added a row to Table 1 with DreamFusion (base) that uses the identical model to Dream Fields (reimpl.) but swaps the CLIP loss for the SDS loss (this is the same base model presented in Fig. 6). The DreamFusion (base) model yields better CLIP Color R-Precision scores than Dream Fields, likely due to the larger dataset and capacity of the Imagen text-to-image model vs. CLIP-B/16 that the Dream Fields (reimpl.) model is trained with.
>
> Are there any other ablations you would like to see included? Running each ablation is costly as we have to optimize 153 NeRF models to compute the CLIP R-Precision on the object-centric COCO validation set. We chose the ordering of the components for our ablations based on the way they were introduced to solve previous problems (e.g. textureless rendering was introduced to correct geometry in dark areas where illuminated renders are insensitive).

---

### Official Review · Reviewer_44Jy · 2022-11-04

**Confidence:** 4
**Correctness:** 4
**Technical Novelty And Significance:** 4
**Empirical Novelty And Significance:** 3
**Recommendation:** 8

**Clarity, Quality, Novelty And Reproducibility:**

The paper writing is clear and easy to follow. The proposed method is also novel and can be an interesting contribution to the diffusion and 3D synthesis community. Also, the algorithm is clear and easy to implement. However, since it relies on a strong pre-trained text-to-image diffusion model, it will be difficult to reproduce the same level of quality without access to the original model.


**Strength And Weaknesses:**

Strength: The method is pretty novel and effective. The analysis of the instability of the loss function is technically sound, and the solution makes the model effective in optimizing the high-quality 3D content based on the text prompt.

Weakness:
(1) the proposed text-to-3D generation is a bit tricky to evaluate. Since the model is per-scene optimization with a pretrained DM and text prompt, both the quality and text input will affect the synthesized output. Also, as it is difficult to have real 3D ground truth to compare the generation quality (in comparison to 2D generation)
(2) while the generated results correspond to the text input well and sometimes generalize to very novel scenes, I think it is only thanks to the original power 2D diffusion model. The generated 3D, however, seems to have severe mode collapse issues compared to 2D models (blurry, dark, and unnatural high-contrast colors). Is it because the objective function encourages that? What might be the possible way to avoid such a generation further? How can we generate multiple 3D scenes with the same text?
(3) the model seems to require a lot of computational resources to optimize for a single scene. Will the same approach be generalized to synthesize the radiance field without optimization directly?


**Summary Of The Paper:**

This paper proposes a novel approach for text-to-3D generation based on pre-trained text-conditioned diffusion models (Imagen in this paper). More specifically, the authors learn a neural radiance field by rendering at random views and use the pretrained diffusion model as the objective function to back-prop gradients to the radiance field. The authors also analyzed the difficulties of directly applying the denoising objective as the loss function and proposed a "simpler" form related to "probability density distillation."

**Summary Of The Review:**

This submission creates a novel way of generating 3D content without needing 3D or 2D image data. Compared to similar existing work (e.g., dreamfield),  the usage of the diffusion model makes it possible to generate highly realistic samples. Also, the proposed fix of using "probability density distillation" loss for learning with a pretrained DM is technically sound, and also has great potential to be applied to other distillation tasks. Therefore, despite some possible weaknesses listed above, it should be a solid to be appear at ICLR.

---

> ### Author Response · Authors · 2022-11-10
> **Response to Reviewer 44Jy**
>
> Thank you for your time and helpful feedback. We respond below to your questions and concerns:
>
> >  the proposed text-to-3D generation is a bit tricky to evaluate
>
> We agree that evaluation of these generative systems is challenging. We make use of the same CLIP-based evaluation framework as prior work in Dream Fields, though we found that it failed to capture the quality of underlying geometry, which led us to introduce an additional evaluation approach on textureless renders. We agree that the lack of real ground-truth 3D data is problematic for 3D generative models (indeed, this observation motivated us to design our method around a 2D generative model, where this problem can be circumvented). The question of designing good metrics for measuring success in 3D generative media appears to be an open research problem, and we are hopeful that our work will encourage others to investigate better ways to empirically measure success. That being said, we feel that the qualitative improvement demonstrated by DreamFusion relative to prior work is highly evident upon visual inspection of the video results in the supplemental website, https://dreamfusionpaper.github.io.
>
> > mode collapse
>
> While there is some diversity across random initializations, we do find that the mode-seeking nature of the loss function results in reduced diversity per text prompt. To encourage modes to be high quality samples, we require high guidance weights which can cause over-saturation and a cartoon-y appearance in the outputs. We have partially ameliorated these artifacts by modifying the prompt (adding “a DSLR photo of”) but agree that this is a limitation of the current approach. And indeed, as the reviewer notes, this mode-seeking tendency is exacerbated when optimizing over a 3D NeRF instead of a 2D image. This is likely due to the complicated dynamics of NeRF optimization, which are a consequence of the inherently difficult and underconstrained nature of 3D reconstruction. However, as we show, users can achieve a great diversity in styles by varying the text.
>
> > computational resources
>
> We agree that text-to-3D without per-scene optimization would be an exciting future direction. We believe it may be possible to amortize our per-text optimization problem across scenes and thereby distill it into a fast feedforward text-to-3D model, and we hope that our existing framework and loss function will help enable this in the future.

---

> > ### Comment · Reviewer_44Jy · 2022-12-01
> > **Thank you for the response!**
> >
> > I am satisfied with the author's response, and it answers my questions.
> > I will keep my score for recommending an acceptance.

---

### Comment · Program_Chairs · 2023-03-21
**Congratulations on winning the ICLR 2023 Outstanding Paper Award!!**


Hello authors,

Big congratulations! For more details about the award this year, please see https://blog.iclr.cc/2023/03/21/announcing-the-iclr-2023-outstanding-paper-award-recipients/

Best,

Mengdi, on behalf of ICLR 2023 PCs

---

### Decision · Program_Chairs · 2023-01-20

**Decision:**

Accept: notable-top-5%

**Justification For Why Not Higher Score:**

N/A

**Justification For Why Not Lower Score:**

This is a well presented paper, introducing a sound algorithm to perform text-to-3D synthesis. The quality of the execution is high and the presented results are solid and qualitatively impressive. The work has the potential to unlock interesting future research.

**Metareview: Summary, Strengths And Weaknesses:**

This paper was reviewed by four knowledgeable referees. Initial reviews suggested that the paper tackles a relevant problem (text-to-3D), the proposed approach is sound, and the quality of the execution high. Reviewers only raised some minor concerns wrt the challenges in evaluating such models (44Jy), the resources required for optimization (44Jy), the positioning of the approach wrt DreamFields (nnv7), and some missing ablations (UY2y). The authors engaged with the reviewers during the discussion phase and addressed most of their concerns. After discussion, reviewers are convinced about the value of the contributions in this paper, and their potential to unlock future research. The AC agrees with the reviewers' assessment and therefore recommends to accept.

**Note From Pc:**

if the above contains the word "oral" or "spotlight" please see: "oral" presentation means -> notable-top-5% and "spotlight" means -> notable-top-25%. As stated in our emails, we are disassociating presentation type from AC recommendations